**Data Availability Statement:** All data used in this study are available at Gene Expression Omnibus, https://www.ncbi.nlm.nih.gov/geo/.

# Bioinformatic analysis identified novel candidate genes with the potentials for diagnostic blood testing of primary biliary cholangitis

Hoang Nam Pham[1], Linh Pham[2], Keisaku Sato[3]*

1 Department of Life Sciences, University of Science and Technology of Hanoi, Hanoi, Vietnam, 2 Department of Science and Mathematics, Texas A&M University–Central Texas, Killeen, Texas, United States of America, 3 Department of Medicine, Division of Gastroenterology and Hepatology, Indiana University School of Medicine, Indianapolis, Indiana, United States of America

* keisato@iu.edu

## Abstract

Primary biliary cholangitis (PBC) is an autoimmune disorder characterized by intrahepatic bile duct destruction and cholestatic liver injury. Diagnosis of PBC is generally based on the existence of anti-mitochondrial antibody (AMA) in blood samples; however, some PBC patients are negative for serum AMA tests, and invasive liver histological testing is required in rare PBC cases. The current study seeks novel candidate genes that are associated with PBC status and have potentials for blood diagnostic testing. Human transcriptomic profiling data of liver and blood samples were obtained from Gene Expression Omnibus (GEO). Three GEO data series (GSE79850, GSE159676, and GSE119600) were downloaded, and bioinformatic analyses were performed. Various differentially expressed genes were identified in three data series by comparing PBC patients and control individuals. Twelve candidate genes were identified, which were upregulated in both liver tissues and blood samples of PBC patients in all three data series. The enrichment analysis demonstrated that 8 out of 12 candidate genes were associated with biological functions, which were closely related to autoimmune diseases including PBC. Candidate genes, especially ITGAL showed good potentials to distinguish PBC with other diseases. These candidate genes could be useful for diagnostic blood testing of PBC, although further clinical studies are required to evaluate their potentials as diagnostic biomarkers.

## Introduction

Primary biliary cholangitis (PBC) is an autoimmune cholestatic liver disease characterized by destruction of intrahepatic bile ducts leading to cholestasis, biliary inflammation, and liver fibrosis [1,2]. PBC is a progressive disorder, and advanced PBC is associated with liver cirrhosis, liver failure, and death [3]. Ursodeoxycholic acid (UDCA) is the FDA-approved medication for PBC, which improves biochemical tests and transplant-free survival [4]. Combination

**Funding:** The authors received no specific funding for this work.

**Competing interests:** The authors have declared that no competing interests exist.

**Abbreviations:** AIH, = autoimmune hepatitis; ALP, alkaline phosphatase; AMA, anti-mitochondrial antibody; ANA, anti-nuclear antibody; AUC, area under the ROC curve; CD, Crohn's disease; DEGs, differentially expressed genes; FDR, false discovery rate; GEO, Gene Expression Omnibus; KEGG, Kyoto Encyclopedia of Genes and Genomes; MCODE, Molecular Complex Detection; NASH, non-alcoholic steatohepatitis; PBC, primary biliary cholangitis; PPI, protein-protein interaction; PSC, primary sclerosing cholangitis; ROC, receiver operating characteristic; SLE, = systemic lupus erythematosus; UC, ulcerative coltis; UDCA, ursodeoxycholic acid.

of UDCA with bezafibrate enhances the therapeutic effects of UDCA, and obeticholic acid also improves transplant-free survival in PBC patients [5,6].

However, PBC can progress quickly, and the median time to develop liver fibrosis is 2 years without any therapeutic treatments [7]. Advanced PBC results in severe liver failure or liver cirrhosis, and liver transplantation is the only treatment option for patients at the end-stage PBC; therefore, early diagnosis is critical to start treatments and avoid liver transplantation [8]. The diagnosis of PBC is based on two features of the disorder: (1) abnormally elevated alkaline phosphatase (ALP) levels indicating cholestasis; and (2) serological reactivity to anti-mito-chondrial antibody (AMA) or anti-nuclear antibody (ANA) indicating autoimmune response [3]. Although AMA tests are the gold-standard method for PBC diagnosis and over 90% PBC patients show positive serum AMA, small percentage of PBC patients does not have serum AMA [9]. Since PBC is caused by the autoimmune response, some patients show autoimmune features and characteristics of autoimmune hepatitis (AIH) [3]. If diagnostic laboratory tests are inconclusive, or AIH or other disorders are suspected, liver biopsy is required for histological analysis to identify bile duct destruction, peribiliary inflammation, or liver fibrosis to confirm PBC status [10].

Although PBC is a relatively rare cholangiopathy, cases are reported predominantly in middle-aged women, and incidence has been increasing from 5.0 in 1987 to 34.6 in 2014 per 100,000 population [11,12]. There is a need for novel PBC biomarkers, which can be used for early diagnostic testing and avoid invasive liver biopsy analysis [13]. The current study aims to identify candidate genes associated with PBC and evaluate their efficacy and potentials to distinguish PBC status from healthy conditions or other diseases.

## Materials and methods

### Data collection

Three sets of expression profiling data for human patients were obtained from Gene Expression Omnibus: GSE79850, GSE159676, and GSE119600 (GEO, https://www.ncbi.nlm.nih.gov/geo/). GSE79850 contains profiling data for liver samples of "low risk" PBC patients (n = 7), who responded fully to UDCA, "high risk" PBC patients (n = 9), who required liver transplantation, and non-diseased control (n = 8) [14]. All 24 samples were analyzed in this study. GSE159676 contains microarray data of liver tissues for controls (n = 6), patients with non-alcoholic steatohepatitis (NASH, n = 7), primary sclerosing cholangitis (PSC, n = 12), PBC (n = 3), AIH (n = 3), haemochromatosis (n = 1), and alcoholic liver disease (n = 1) [15]. Data of haemochromatosis and alcoholic liver disease were excluded in this study due to insufficient sample numbers. GSE119600 contains microarray data for whole blood samples of controls (n = 47), patients with PBC (n = 90), PSC (n = 45), Crohn's disease (CD, n = 95), and ulcerative coltis (UC, n = 93) [16]. All 370 samples were included in this study. All data did not contain information of patients, and the authors could not identify individual participants during and after data collection.

### Data processing

R version 4.2.2 was used in this study. Normalized code counts, quantile normalized data, and quantile normalized signal intensity for GSE79850, GSE159676, and GSE119600, respectively, were collected from series matrix files using the GEOquery package [17]. Probes were annotated based on information provided in data files. All data were converted to log2 values for further analysis. We identified outstanding outliers in data series, which were over 100-fold higher probe values compared to other values for the same probe, using the boxplot function in base R graphics (range = 10). GSE159676 contained 4,829 outliers, and they were replaced

by NA (not applicable) to perform powerful and accurate statistical analysis. Percentage of replaced data values with NA was under 1% of total data value numbers. Qualities of data to be analyzed after outlier replacement were checked by the boxplot function for log2 values to see data distribution (S1 Fig). Outstanding outliers were not identified in GSE79850 and GSE119600, and all data points of these data series were used for further analysis. For GSE79850, "high risk" and "low risk" PBC patients were merged in one group as "PBC" group.

### Identification of differentially expressed genes (DEGs)

DEGs comparing control and PBC groups in three data series were identified using the limma package [18]. Volcano plots were drawn using the ggplot2 package [19]. Adjusted p values were calculated by the false discovery rate (FDR) [20]. DEGs with adjusted P<0.05 were considered as statistically significant. DEGs were sorted according to the B-statistic, and top 100 DEGs were plotted in heatmaps using the pHeatmap package. When there were multiple probes for the same gene in top 100 DEGs, the probe with the highest B-statistic was selected and others were not used to draw heatmaps for that gene. For GSE159676, some probe data with outstanding outliers (replaced by NA) were removed from top 100 DEGs, since pHeatmap cannot handle NA data points.

### Identification of candidate genes and networks

Venn diagrams were drawn using the VennDiagram package [21], and the list of genes at the intersection was obtained using the venn function in the gplots package. Function enrichment analysis on Kyoto Encyclopedia of Genes and Genomes (KEGG) was performed using the clusterProfiler package [22] to understand the biological functions for candidate genes at the intersection. Protein-protein interaction (PPI) networks for candidate targets were identified using stringApp (version 2.0.1) [23] in Cytoscape software (version 3.9.1) [24]. The molecular complex in identified PPI was identified using Molecular Complex Detection (MCODE) in Cytoscape [25].

### Validation of candidate genes

The expression levels of identified candidate genes were validated in three data series. Normalized probe counts or values were used to compare the control, PBC, and other disorders. Probe data were processed by the original authors of each data series for normalization, and no further data process was performed. For GSE79850, low risk and high risk PBC patients were analyzed separately. Unpaired Student's *t*-test was performed and bar graphs were drawn using Prism software (version 9.5.1). The pROC package was used to calculate receiver operating characteristic (ROC) curve [26]. Area under the ROC curve (AUC), sensitivity, and specificity were recorded. ROC curves were drawn using the ggplot2 package [19].

## Results and discussion

### Identification of upregulated genes in PBC patients

Transcriptomic profiling data were analyzed between controls and PBC patients in GSE79850 (liver tissues, 16 PBC samples and 8 control samples), GSE159676 (liver tissues, 3 PBC samples and 6 control samples), and GSE119600 (whole blood, 90 PBC samples and 47 control samples). Total numbers of probes included in this study are 779 for GSE79850, 17,046 for GSE159676, and 47,230 for GSE119600. A total of 241 DEGs were identified in GSE79850 (66 downregulated genes and 175 upregulated genes), 2,695 DEGs in GSE159676 (1,430 downregulated and 1,265 upregulated), and 5,707 DEGs in GSE119600 (2,874 downregulated and

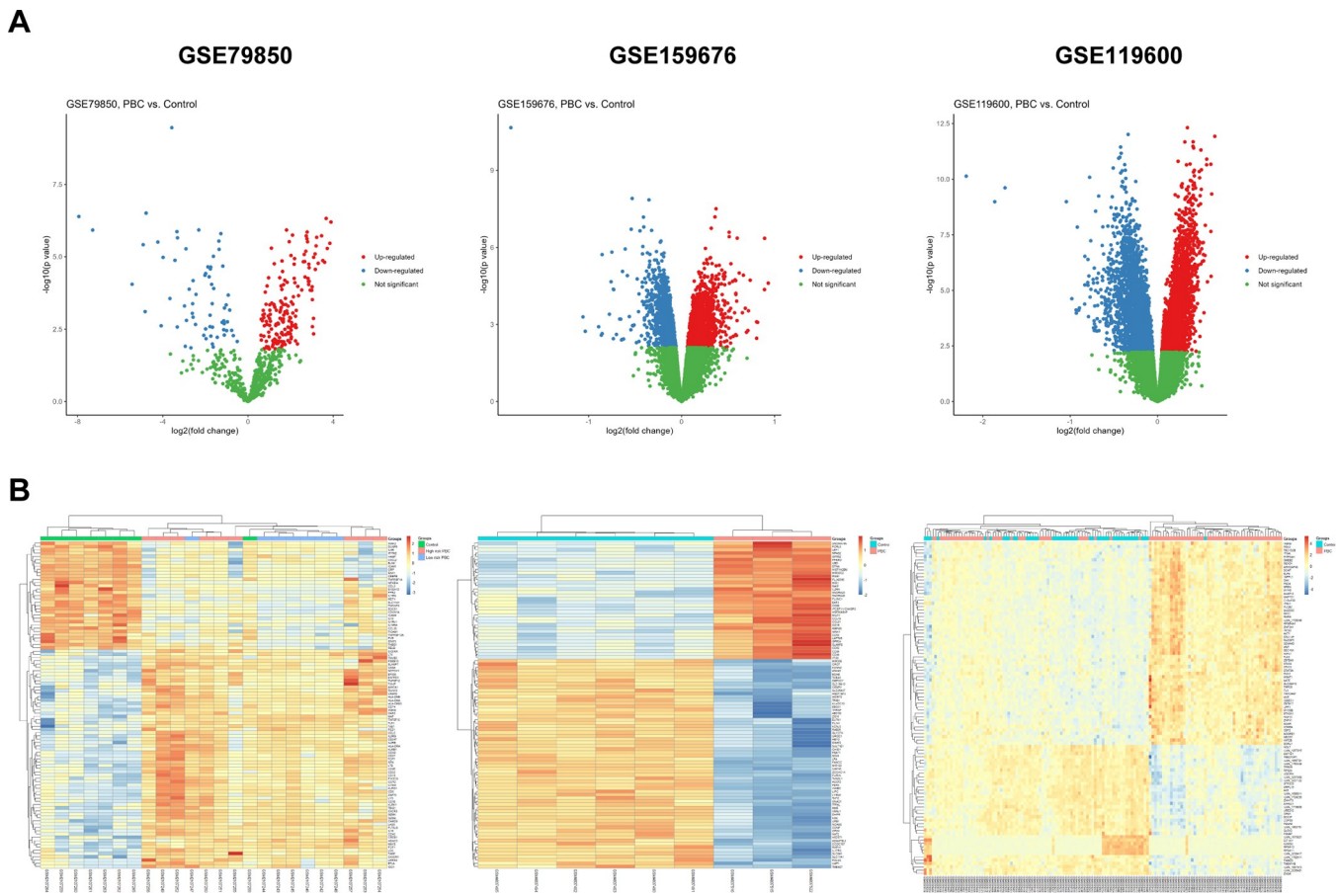

**Fig 1. Identification and plots of DEGs in PBC patients.** (A) Volcano plots of DEGs identified in three data series. (B) Heatmaps of top 100 DEGs comparing PBC and control samples.

2,833 upregulated) including duplicated probes or probes without annotations (Fig 1A). Top 100 DEGs identified in three data series are shown in heatmaps (Fig 1B).

## Enrichment analysis and identification of PPI networks for candidate genes

The venn diagrams did not show downregulated genes at the intersection of three data series; however, they identified 12 genes that were upregulated across all three data series (Fig 2A): BTK, CD44, FYN, IDO1, IKBKB, IL21R, INPP5D, ITGA4, ITGAL, PIK3CG, PRKCD, and SYK. KEGG enrichment analysis identified 5 biological functions/pathways (FcεRI signaling, B cell receptor signaling, Epstein-Barr virus infection, platelet activation, and osteoclast differentiation) associated with 8 genes out of 12 candidate genes (Fig 2B). Previous studies have shown that Fc receptors, including FcεRI, are associated with autoimmunity [27]. B cell signaling contributes to autoimmunity [28], and altered functions of B cells are associated with PBC [29,30]. It is known that Epstein-Barr virus infection induces autoimmune liver diseases including PBC [31,32]. Recent studies have identified elevated platelet activation in early-stage PBC patients [33,34]. Patients with PBC have increased risks of osteoporosis, which is a common complication in PBC [35,36], and pathways in osteoclast differentiation are associated with autoimmune diseases and PBC [37,38]. These previous studies and our findings indicate

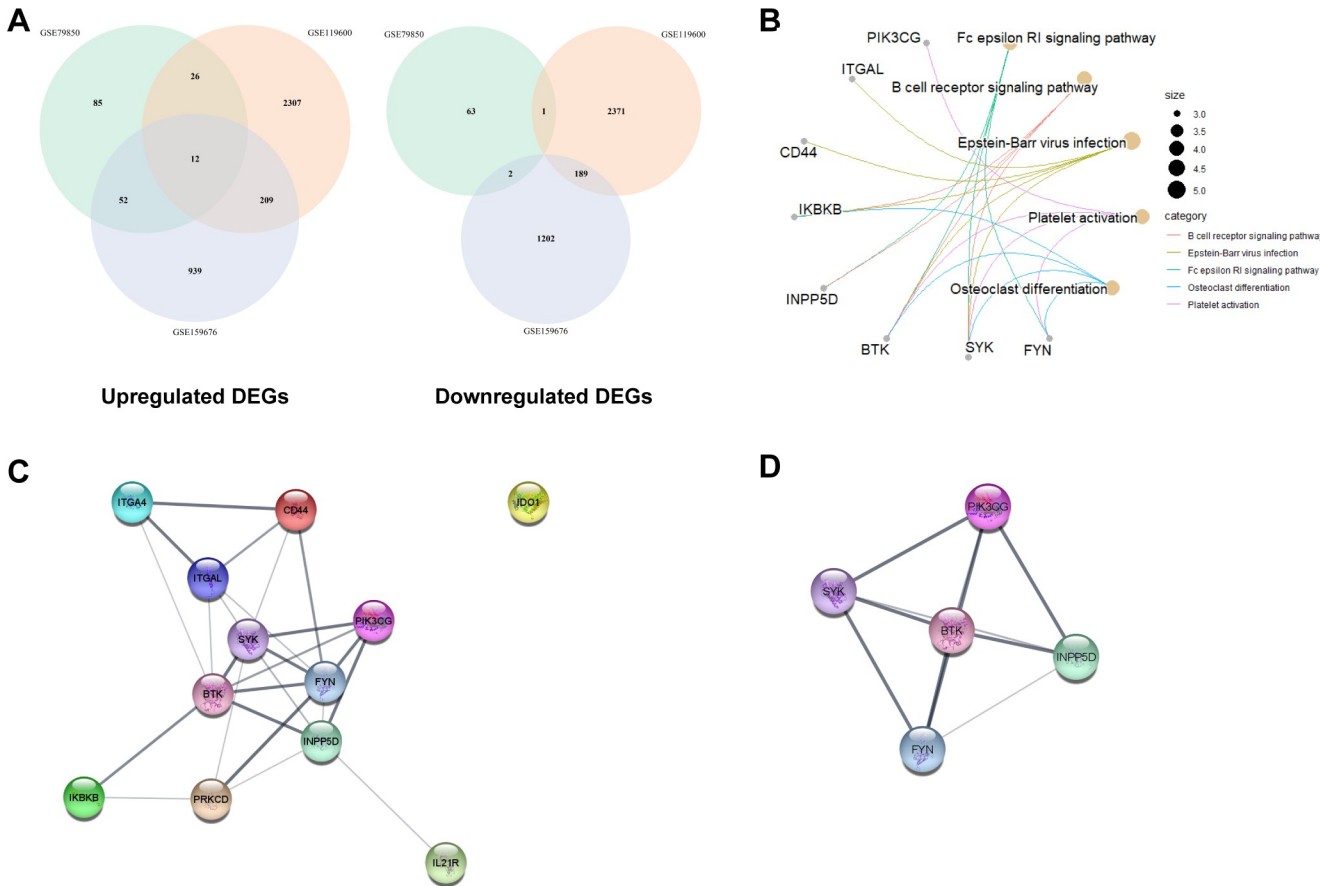

**Fig 2. The association of 12 candidate genes with PBC.** (A) Venn diagrams for upregulated and downregulated DEGs comparing three data series. No genes were identified, which were downregulated in all three data series. (B) KEGG enrichment analysis for 12 candidate genes. The association was identified for 8 out of 12 candidate genes. (C) PPI network analysis. All candidate genes except IDO1 showed interaction with others. (D) Suggested molecular complex within the identified PPI network using MCODE.

that 5 biological functions identified are closely associated with PBC, and 8 genes in these 5 functions (BTK, CD44, FYN, IKBKB, INPP5D, ITGAL, PIK3CG, and SYK) could be critical in PBC. STRING network analysis recognized the PPI network in 12 candidate genes, except IDO1 (Fig 2C). MCODE identified the molecular complex in this network, which consists of BTK-FYN-INPP5D-PIK3CG-SYK (Fig 2D). These results support that candidate genes, especially 8 genes, are closely related to PBC.

## Validation of candidate genes

Expression levels of all 12 candidate genes were validated in three data series using normalized count values (Figs 3 and S2). The key genes identified in KEGG enrichment analysis, such as ITGAL, and those in the molecular complex, such as FYN, IKBKB, and INPP5D, showed significantly elevated expression levels in both liver tissues and whole blood samples of PBC patients compared to controls. Expression levels of these genes, especially ITGAL, in PBC patients were significantly higher than patients with other diseases, such as PSC, NASH, and AIH in liver samples and PSC, CD, and UC in blood samples. Interestingly, KEGG enrichment analysis or PPI network analysis did not identify the link of IDO1 with other candidate genes (Fig 2), but IDO1 expression was significantly elevated in PBC patients compared to control

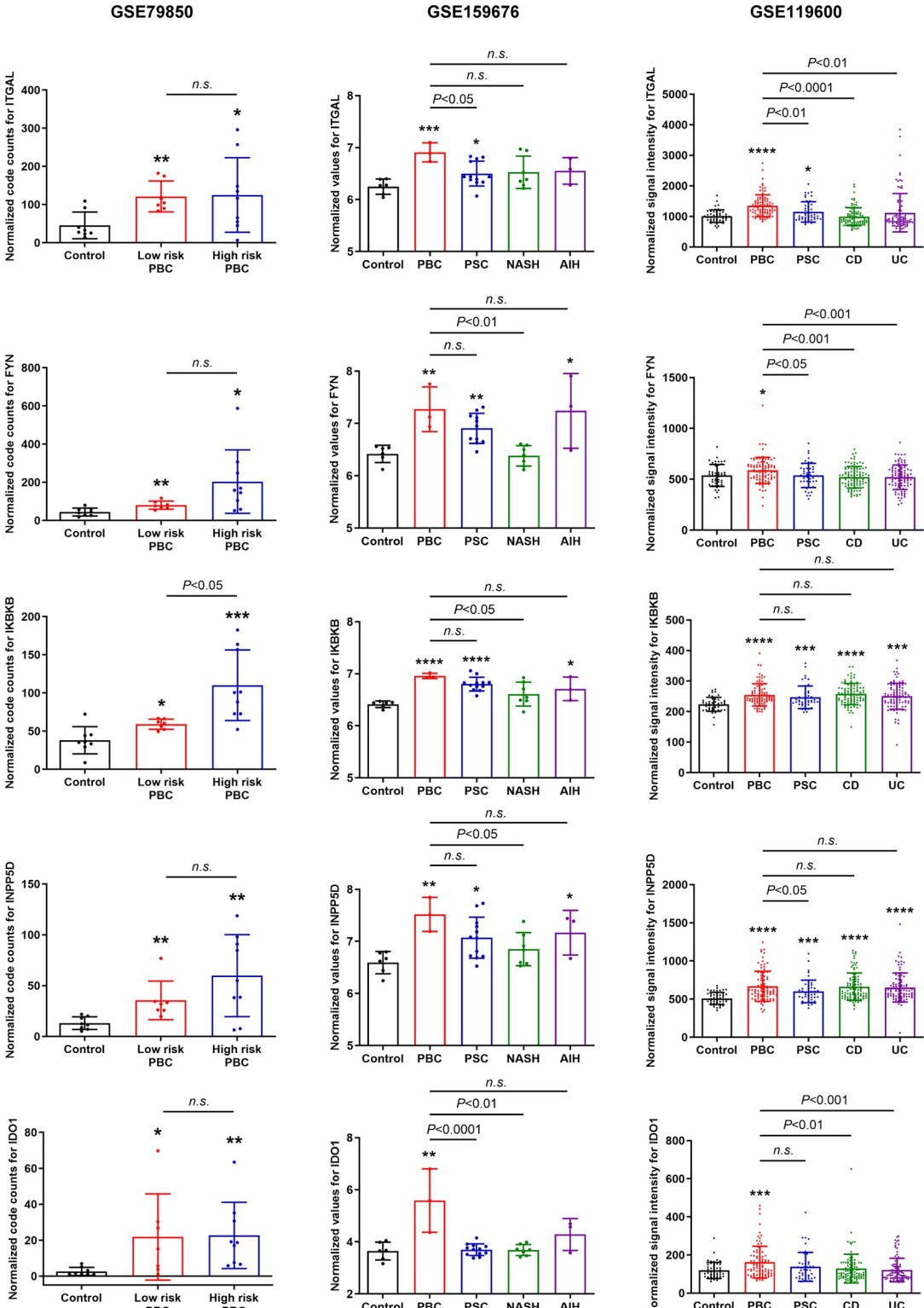

**Fig 3. Validation of selected candidate gene expression in each data series.** Normalized code counts, quantile normalized data, and quantile normalized signal intensity (non-log2 values) for GSE79850, GSE159676, and GSE119600, respectively, were plotted, and unpaired Student's *t*-test was performed. Mean±SD, *P<0.05, **P<0.01, ***P<0.001, and ****P<0.0001 compared to the Control group.

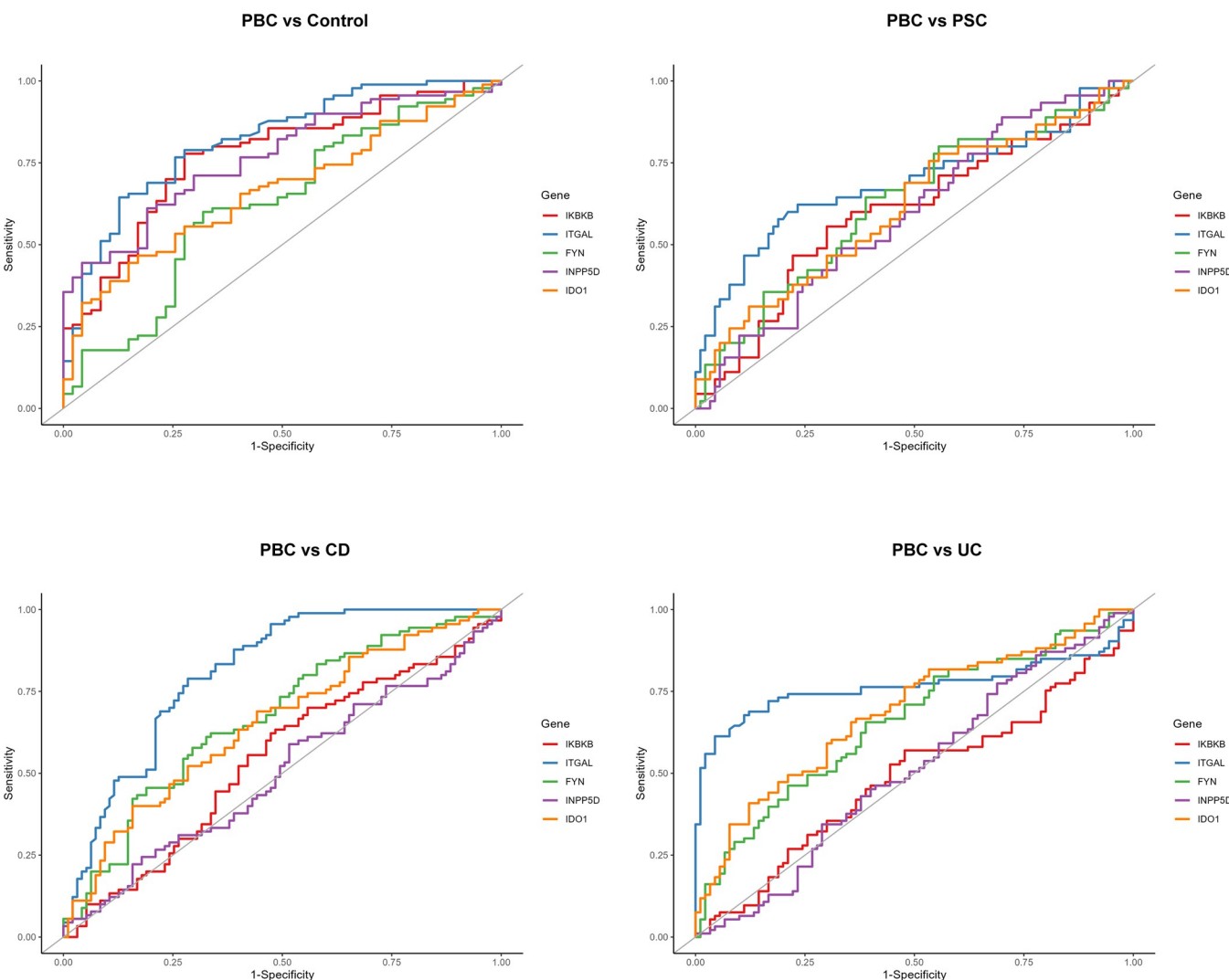

**Fig 4. ITGAL showed the best potential to distinguish PBC from other diseases.** ROC curves for selected candidate genes to distinguish PBC from control or other disorders in GSE119600.

individuals and patients with other disorders (Fig 3). The data series GSE119600 has whole blood transcriptomic profiling data for total 370 samples, including controls, PBC, PSC, CD, and UC [16]. To evaluate the potentials of 12 candidate genes to distinguish PBC status from healthy conditions and other diseases by blood testing, ROC curve analysis was performed (Fig 4). ITGAL showed highest AUC and showed diagnostic potentials not only to distinguish PBC status from healthy controls, but also from PSC, CD, and UC (Fig 4 and Table 1). Other candidate genes did not show such potentials as ITGAL (Tables 1 and S1). ROC curve analysis was also performed using data series GSE79850 and GSE159676, and all 12 genes showed high AUC; however, these data series have fewer sample numbers and data are not as strong as those using GSE119600 (S3 and S4 Figs, S2 and S3 Tables).

## Significance, limitations and future perspectives

The current study identified candidate genes associated with PBC. Validation using three data series strongly support that expression levels of the 12 candidate genes, BTK, CD44, FYN,

**Table 1. ROC curve data of GSE119600 for selected candidate genes.**

| | PBC vs Control | | |
| --- | --- | --- | --- |
| | AUC | Sensitivity (%) | Specificity (%) |
| **FYN** | 0.6241 | 60 | 68.1 |
| **IDO1** | 0.6652 | 55.6 | 72.3 |
| **IKBKB** | 0.7721 | 77.8 | 72.3 |
| **INPP5D** | 0.7674 | 71.1 | 70.2 |
| **ITGAL** | 0.8197 | 76.7 | 74.5 |
| | PBC vs PSC | | |
| | AUC | Sensitivity (%) | Specificity (%) |
| **FYN** | 0.6264 | 64.4 | 61.1 |
| **IDO1** | 0.6146 | 68.9 | 52.2 |
| **IKBKB** | 0.5985 | 60 | 55.6 |
| **INPP5D** | 0.5956 | 48.9 | 66.7 |
| **ITGAL** | 0.69 | 62.2 | 76.7 |
| | PBC vs CD | | |
| | AUC | Sensitivity (%) | Specificity (%) |
| **FYN** | 0.6701 | 62.2 | 66.3 |
| **IDO1** | 0.6516 | 68.9 | 55.8 |
| **IKBKB** | 0.5427 | 62.2 | 0.526 |
| **INPP5D** | 0.5019 | 58.9 | 48.4 |
| **ITGAL** | 0.8102 | 78.9 | 71.6 |
| | PBC vs UC | | |
| | AUC | Sensitivity (%) | Specificity (%) |
| **FYN** | 0.6575 | 65.6 | 61.1 |
| **IDO1** | 0.6863 | 65.6 | 64.4 |
| **IKBKB** | 0.4858 | 57.0 | 52.2 |
| **INPP5D** | 0.5098 | 46.2 | 58.9 |
| **ITGAL** | 0.7658 | 72.0 | 83.3 |

IDO1, IKBKB, IL21R, INPP5D, ITGA4, ITGAL, PIK3CG, PRKCD, and SYK, are significantly upregulated in whole blood and liver tissues of PBC patients. ITGAL was identified in KEGG enrichment analysis showing the link with Epstein-Barr virus infection, which is associated with PBC [31,32]. Elevated ITGAL expression could be useful to diagnose PBC and distinguish this disorder from other diseases, such as PSC, CD, and UC.

Blood tests for PBC patients are performed to detect AMA, and approximately 95% PBC patients have AMA in blood samples [39]. A previous study testing 4 commercial assay kits for AMA serologic tests to distinguish PBC patients from patients with other liver diseases showed sensitivity of 55.7–79.7% and specificity of 91.7–95.4% [40]. Blood profiling data for ITGAL in GSE119600 showed sensitivity of 62.2–78.9% and specificity of 71.6–83.3% in distinguishing PBC from controls, PSC, CD, or UC. Expression analysis for ITGAL using blood samples may be beneficial for patients with PBC who do not exhibit serum AMA or display characteristics of other diseases. This approach can offer an alternative to invasive liver tissue analysis for PBC diagnosis.

ITGAL is the integrin alpha L chain, and integrin is associated with autoimmunity [41]. Both ITGAL and ITGA4 (integrin subunit alpha 4) were identified as two of the 12 candidate genes in the current study and may be associated with the pathophysiology of PBC. However, functional roles of ITGA4 and ITGAL in PBC are largely unknown, and further studies are required to elucidate the detailed mechanisms of ITGAL-mediated autoimmunity and PBC. A

previous study demonstrated that infiltrated lymphocytes express elevated ITGA4 in liver samples of patients with PBC and chronic viral hepatitis [42]. It is well known that altered functions of T lymphocytes are involved in the pathophysiology of PBC [43,44]. Elevated expression of integrin αvβ6 was identified in biliary cells of liver sections from PBC and PSC patients [45]. Ductular reaction and liver fibrosis are a hallmark in PBC and PSC, and elevated ductular reaction is closely associated with hepatic fibrogenesis [46,47]. Integrin αvβ5/αvβ3 mediates NF-κB activation leading to ductular reaction during cholestatic liver injury [48]. Future studies may elucidate the pathological roles of ITGAL in PBC via integrin.

Other candidate genes may be associated with the pathogenesis of PBC. For example, IDO1 (indoleamine 2,3-dioxygenase 1) is expressed in cholangiocytes and hepatocytes of PBC liver sections [49]. Tryptophan to kynurenine conversion was increased in PBC patients compared to control individuals, indicating elevated enzymatic IDO1 activity during PBC [49]. Another study also identified elevated IDO1 expression in cholangiocytes of PBC patients compared to controls [50]. Although nothing was revealed in KEGG enrichment analysis and PPI network analysis for IDO1 in the current study, IDO1 expression was significantly elevated in PBC patients' liver tissues and blood samples, indicating the potential association of IDO1 with the pathophysiology of PBC. Another candidate gene includes IL21R, which is the IL-21 receptor. IL-21 is a cytokine produced by CD4[+] T cells, and studies indicate that abnormal population of CD4[+] T cells and upregulation of IL-21R signaling may play an important role in autoimmune disorders, such as systemic lupus erythematosus (SLE) [51,52]. Although a previous study suggested the potential association of IL-21 with PBC [53], functional roles of the IL-21/IL-21R axis in PBC and autoimmune disorders are still undefined. Further studies are required to conclude the association with PBC and elucidate functional roles of IL-21 signaling.

The current study has multiple limitations: (1) all three data series do not contain clinical data (e.g., serum ALT levels or fibrosis levels) or follow-up information of patients (e.g., liver fibrosis progression or required liver transplantation). Therefore, the correlation between candidate genes with liver pathology is unclear (e.g., ITGAL expression is positively correlated with liver fibrosis levels). Further clinical studies are required to identify the correlation and establish the link between candidate genes and PBC; (2) sample numbers are relatively small, especially for GSE159676. Comparing PBC (n = 3) to AIH (n = 3) of GSE159676 was not statistically significant in all cases because of small sample numbers (Figs 3 and S2). Liver biopsy tests are required for PBC patients if co-existence of NASH or AIH is suspected [3]. Expression levels of ITGAL in liver samples of GSE159676 were not significantly higher than NASH (n = 7) or AIH, due to small sample numbers (Fig 3). ROC curve analysis was not powerful with small sample size in GSE159676 (S4 Fig); (3) GSE119600 does not include NASH or AIH patients, and further studies are required to determine if blood expression tests for ITGAL can distinguish PBC from NASH or AIH; (4) sample collection and experimental procedures are not consistent in three data series. Control blood samples of GSE119600 were obtained from healthy individuals [16], but control liver samples of GSE159676 were from patients with colorectal cancer, not healthy subjects [15]. Microarray data could differ depending on methods of sample collection or the location of collected samples in the liver tissues (e.g., liver zonation [54]). Procedures of RNA isolation and microarray platforms are different in all three data series. While the current study has provided valuable insights with consistent findings regarding for ITGAL and other candidate genes across three data series, it is important to acknowledge that diverse cohorts, sample collections, or procedures may yield varying outcome; and (5) clinical studies are required to evaluate the potentials of candidate genes as diagnostic PBC biomarkers. Our study is based on publicly available database and computation, and clinical data are essential to establish a novel blood testing method for ITGAL to diagnose PBC.

## Conclusions

Our analysis using GEO data series identified 12 candidate genes associated with PBC, most notably ITGAL, which could be useful for future diagnostic blood testing. Clinical studies are required to evaluate the capability of ITGAL as a diagnostic biomarker for PBC, and further experimental evidence is needed to elucidate the roles of ITGAL in the pathophysiology of PBC.

## Supporting information

**S1 Fig. Box plots for log2 data values of GSE159676 before and after outlier replacement with NA.**
(PDF)

**S2 Fig. Validation of other candidate gene expression in each data series.** Mean±SD, *$P<0.05$, **$P<0.01$, ***$P<0.001$, and ****$P<0.0001$ compared to the Control group.
(PDF)

**S3 Fig. ROC curves for selected candidate genes to distinguish PBC from control or high and low risk PBC in GSE79850.**
(PDF)

**S4 Fig. ROC curves for selected candidate genes to distinguish PBC from control or other disorders in GSE159676.**
(PDF)

**S1 Table. ROC curve data of GSE119600 for 12 candidate genes.**
(DOCX)

**S2 Table. ROC curve data of GSE79850 for 12 candidate genes.**
(DOCX)

**S3 Table. ROC curve data of GSE159676 for 12 candidate genes.**
(DOCX)

## Author Contributions

**Conceptualization:** Keisaku Sato.

**Data curation:** Hoang Nam Pham, Linh Pham, Keisaku Sato.

**Investigation:** Keisaku Sato.

**Methodology:** Keisaku Sato.

**Software:** Keisaku Sato.

**Supervision:** Keisaku Sato.

**Validation:** Keisaku Sato.

**Visualization:** Keisaku Sato.

**Writing – original draft:** Keisaku Sato.

**Writing – review & editing:** Hoang Nam Pham, Linh Pham.

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
