## [Decision Letter · Decision Letter 0]

9 Jun 2023

PONE-D-23-10430ITGAL is a potential biomarker for diagnostic blood testing of primary biliary cholangitisPLOS ONE

Dear Dr. Sato,

Thank you for submitting your manuscript to PLOS ONE. After careful consideration, we feel that it has merit but does not fully meet PLOS ONE’s publication criteria as it currently stands. Therefore, we invite you to submit a revised version of the manuscript that addresses the points raised during the review process.

We look forward to receiving your revised manuscript.

Kind regards,

Antonio De Vincentis

Academic Editor

PLOS ONE

Journal Requirements:

2. Please include your tables as part of your main manuscript and remove the individual files. Please note that supplementary tables (should remain/ be uploaded) as separate "supporting information" files

Additional Editor Comments:

Please note that also Obeticholic acid and Fibrate therapy showed to have a beneficial effect on PBC (doi: 10.1016/j.jhep.2021.04.010 and doi: 10.1053/j.gastro.2022.08.054).

Please update the introduction.

Reviewers' comments:

Reviewer's Responses to Questions

**Comments to the Author**

1. Is the manuscript technically sound, and do the data support the conclusions?

Reviewer #1: Yes

2. Has the statistical analysis been performed appropriately and rigorously? 

Reviewer #1: Yes

3. Have the authors made all data underlying the findings in their manuscript fully available?

Reviewer #1: No

4. Is the manuscript presented in an intelligible fashion and written in standard English?

Reviewer #1: Yes

5. Review Comments to the Author

Reviewer #1: 1. In a section of 2.2 Data processing in page 12, GSE159676 contained 4829 outliers and these were excluded. In Supplementary Figure 1, box plot after excluding outliers showed that the range of gene expression is 2.0 to 3.0. However, in Figure 3, the mean value +SD for ITGAL, FYN, IKBKB, INPP5D, IDO1 are almost 6-7+1-2. These results may indicate that ITGAL, FYN, IKBKB, INPP5D, IDO1 are all outliers. Could you explain more in detail ?

2. In general, needle liver biopsy is not performed in healthy controls. However, the studies in GSE79850 and GSE159676 have data of liver samples derived from healthy subjects.

Will you please describe the method how to obtain liver samples in these studies, since we know that mRNA expression of liver samples markedly differs depending on the method of sample collection.

6. PLOS authors have the option to publish the peer review history of their article (what does this mean?). If published, this will include your full peer review and any attached files.

Reviewer #1: No

---

## [Author Response · Author response to Decision Letter 0]

14 Jul 2023

Comment from the Editor:

Please note that also Obeticholic acid and Fibrate therapy showed to have a beneficial effect on PBC.

Our response:

Thank you for your suggestion. We updated the introduction to introduce studies for obeticholic acid and bezafibrate (refs 5 and 6). We appreciate this comment and for providing us with the reference information.

Comment from the Reviewer:

In a section of 2.2 Data processing in page 12, GSE159676 contained 4829 outliers and these were excluded. In Supplementary Figure 1, box plot after excluding outliers showed that the range of gene expression is 2.0 to 3.0. However, in Figure 3, the mean value +SD for ITGAL, FYN, IKBKB, INPP5D, IDO1 are almost 6-7+1-2. These results may indicate that ITGAL, FYN, IKBKB, INPP5D, IDO1 are all outliers. Could you explain more in detail?

Our response:

Thank you for your comment. In short, the data values look different because the probe count values used for boxplot in Supplementary Figure 1 are log2 values; however, the values used for gene expression validation in Figure 3 and Supplementary Figure 2 are non-log2 values.

It is common to calculate and use log2 values for boxplot because it facilitates the visualization of data distribution in many cases, making it easier to interpret the results. For gene expression validation, we used non-log2 values because we wanted to show gene expression data (i.e., probe read counts) with minimal data processing. There are three data series analyzed in this study; however, each series employ different microarray platforms and data processing techniques (i.e., background subtraction, normalization, or batch correction) performed by the authors of each study. For consistency, we decided to use the original data uploaded in GEO by the authors and use them for gene expression validation in non-log2 formats. We agree with the reviewer that this could be confusing or misleading. We updated the method section and figure legends to provide clarity regarding this difference.

The rationale behind outlier removal is based on whether they are “obviously strange” data points. To define “obviously strange” data points to be removed and replaced by NA, we used boxplot function in R with range 10, as mentioned in the method section. For example, FYN in the GSE159676 data series has one outlier data point. The probe read count for FYN in one PSC patient is 549873 while counts for FYN in other individuals range from 6 to 8. Such a high read count of 549873 clearly indicates an anomaly or error in the data. There are many possibilities to produce these outliers, but we do not know the exact cause because the authors of the data series did not provide any clarification for this in their studies. After outlier removal and replacement, there are still many outliers found (S1 Fig), but they are not “obviously strange”, so we included these data points in our analysis. We checked data of GSE159676, and found that we only removed one outlier data from FYN and one from SYK. No outliers were found for other candidate genes including ITGAL. Outstanding outliers were identified only in GSE159676 data series, and no outliers were removed from data in GSE79850 and GSE119600.

In summary, we replaced 4,829 outliers, which accounts for less than 1% of the total data points (528,426), from GSE159676 using boxplot function in R with range 10. This is the data trimming and cleaning procedure to ensure data integrity rather than to manipulate the data or significantly alter the results. Validation of gene expression using three data series strongly supports the elevated expression of the 12 candidate genes in PBC patients. We updated the manuscript to describe this more clearly. 

Comment from the Reviewer:

In general, needle liver biopsy is not performed in healthy controls. However, the studies in GSE79850 and GSE159676 have data of liver samples derived from healthy subjects. Will you please describe the method how to obtain liver samples in these studies, since we know that mRNA expression of liver samples markedly differs depending on the method of sample collection.

Our response:

Thank you for your comment. We checked the original papers and contacted the authors of the GSE79850 and GSE159676 data series. The GSE159676 authors stated that control liver tissues were obtained from patients with colorectal cancer metastasis, not healthy individuals. These liver tissues were tumor-free and used as control liver tissues, but they were not from healthy individuals. We edited the manuscript for control samples of GSE159676 to describe them as “control liver”, not healthy controls. We could not get any reply from the authors of GSE79850 including the first author and two senior authors after several contacts. In their paper, the formalin fixed paraffin embedded liver tissue samples were obtained from the cellular pathology department archive of Newcastle Hospitals NHS Foundation Trust. It seems that those samples are produced and stored in the organization in UK, and the authors probably do not know if control liver tissues were from healthy individuals or deceased donors or patients with other diseases or how these liver tissues were obtained. The authors of GSE79850 called control samples as “controls”, not “healthy controls” or “healthy individuals”. Control samples in GSE119600 are from healthy individuals because it is clearly described in the paper; however, control liver tissues of GSE79850 and GSE159676 should not be described as healthy controls. We carefully checked our descriptions and edited them correctly. We agree with the reviewer that microarray results may differ depending on the method of sample collection. We updated the section to discuss this.

---

## [Decision Letter · Decision Letter 1]

30 Aug 2023

PONE-D-23-10430R1ITGAL is a potential biomarker for diagnostic blood testing of primary biliary cholangitisPLOS ONE

Dear Dr. Sato,

Thank you for submitting your manuscript to PLOS ONE. After careful consideration, we feel that it has merit but does not fully meet PLOS ONE’s publication criteria as it currently stands. Therefore, we invite you to submit a revised version of the manuscript that addresses the points raised during the review process.

We look forward to receiving your revised manuscript.

Kind regards,

Antonio De Vincentis

Academic Editor

PLOS ONE

Journal Requirements:

Reviewers' comments:

Reviewer's Responses to Questions

**Comments to the Author**

1. If the authors have adequately addressed your comments raised in a previous round of review and you feel that this manuscript is now acceptable for publication, you may indicate that here to bypass the “Comments to the Author” section, enter your conflict of interest statement in the “Confidential to Editor” section, and submit your "Accept" recommendation.

Reviewer #2: All comments have been addressed

2. Is the manuscript technically sound, and do the data support the conclusions?

Reviewer #2: Yes

3. Has the statistical analysis been performed appropriately and rigorously? 

Reviewer #2: Yes

4. Have the authors made all data underlying the findings in their manuscript fully available?

Reviewer #2: Yes

5. Is the manuscript presented in an intelligible fashion and written in standard English?

Reviewer #2: Yes

6. Review Comments to the Author

Reviewer #2: Comments to the author:

This research aimed to compare gene expression differences between PBC patients and controls using GEO data. The authors have finally found 12 gene expressions were significantly different between the two groups. They have also suggested that one of 12 candidates, ITGAL, could be a next biomarker for PBC diagnosis. In clinical setting, new serological diagnostic tools must be needed specifically for AMA seronegative PBC diagnosis. The manuscript is well-written and easy to follow. The study design and study result interpretations are reasonable. Methodologically, the study was properly conducted. The authors have replied to the first reviewer’s comments in the resubmitted version. However, one of concerns is that they have not conducted clinical study to measure ITGAL in the serum for clinical comparisons. Therefore, ITGAL gene expression was different from other patient populations but clinical utility of ITGAL has not been evidenced yet. Indeed, the authors by themselves concluded that our analysis using GEO data series identified 12 novel PBC biomarkers, among which ITGAL stands out as a particularly notable candidate, that could be useful for diagnostic blood testing. Therefore, I would recommend making careful revision of the study title and rephrasing the word of “biomarker” throughout the paper in order to prevent readers from misunderstanding and misleading.

7. PLOS authors have the option to publish the peer review history of their article (what does this mean?). If published, this will include your full peer review and any attached files.

Reviewer #2: **Yes: **SATORU JOSHITA

---

## [Author Response · Author response to Decision Letter 1]

1 Sep 2023

Our response to comments from editors and reviewers is as follows:

Comment from the Reviewer:

The authors have replied to the first reviewer’s comments in the resubmitted version. However, one of concerns is that they have not conducted clinical study to measure ITGAL in the serum for clinical comparisons. Therefore, ITGAL gene expression was different from other patient populations but clinical utility of ITGAL has not been evidenced yet. Indeed, the authors by themselves concluded that our analysis using GEO data series identified 12 novel PBC biomarkers, among which ITGAL stands out as a particularly notable candidate, that could be useful for diagnostic blood testing. Therefore, I would recommend making careful revision of the study title and rephrasing the word of “biomarker” throughout the paper in order to prevent readers from misunderstanding and misleading.

Our response:

Thank you for your comment. We agree with that it could be misleading to say as if ITGAL is a novel biomarker that is useful for PBC diagnosis. We checked and proofread the entire manuscript carefully and changed description from “biomarkers” to “candidate genes” where appropriate. We also updated the abstract, discussion, and conclusion for clarity. All modifications were highlighted in the text. We appreciate your constructive feedback.

---

## [Decision Letter · Decision Letter 2]

4 Oct 2023

Bioinformatic analysis identified novel candidate genes with the potentials for diagnostic blood testing of primary biliary cholangitis

PONE-D-23-10430R2

Dear Dr. Sato,

We’re pleased to inform you that your manuscript has been judged scientifically suitable for publication and will be formally accepted for publication once it meets all outstanding technical requirements.

Kind regards,

Antonio De Vincentis

Academic Editor

PLOS ONE

Additional Editor Comments (optional):

Reviewers' comments:

Reviewer's Responses to Questions

**Comments to the Author**

1. If the authors have adequately addressed your comments raised in a previous round of review and you feel that this manuscript is now acceptable for publication, you may indicate that here to bypass the “Comments to the Author” section, enter your conflict of interest statement in the “Confidential to Editor” section, and submit your "Accept" recommendation.

Reviewer #2: All comments have been addressed

2. Is the manuscript technically sound, and do the data support the conclusions?

Reviewer #2: Yes

3. Has the statistical analysis been performed appropriately and rigorously? 

Reviewer #2: Yes

4. Have the authors made all data underlying the findings in their manuscript fully available?

Reviewer #2: Yes

5. Is the manuscript presented in an intelligible fashion and written in standard English?

Reviewer #2: Yes

6. Review Comments to the Author

Reviewer #2: Comments to the author:

I would very much appreciate the author’s responses. The revised paper was updated with the authors’ comments and amendments. Thank you for your courteous intervention.

7. PLOS authors have the option to publish the peer review history of their article (what does this mean?). If published, this will include your full peer review and any attached files.

Reviewer #2: **Yes: **Satoru Joshita

---

## [Editor Report · Acceptance letter]

6 Oct 2023

PONE-D-23-10430R2 

Bioinformatic analysis identified novel candidate genes with the potentials for diagnostic blood testing of primary biliary cholangitis 

Dear Dr. Sato:

I'm pleased to inform you that your manuscript has been deemed suitable for publication in PLOS ONE. Congratulations! Your manuscript is now with our production department. 

Kind regards, 

on behalf of

Dr. Antonio De Vincentis 

Academic Editor

PLOS ONE